# Study on Solidification Process and Residual Stress of SiCp/Al Composites in EDM

**DOI:** 10.3390/mi13060972

**Published:** 2022-06-19

**Authors:** Wenchao Zhang, Hao Chang, Yu Liu

**Affiliations:** 1School of Mechanical Engineering and Automation, Dalian Polytechnic University, Dalian 116039, China; traum525@gmail.com; 2Department of Mechanical Engineering, Zhengzhou Technical College, Zhengzhou 450121, China; changhaosc@163.com; 3School of Mechanical Engineering, Dalian Jiaotong University, Dalian 116028, China

**Keywords:** EDM, SiCp/Al, recast layer, residual stress, solidification

## Abstract

To study the change of residual stress during heating and solidification of SiCp/Al composites, a one-way FSI (Fluid Structure Interaction) model for the solidification process of the molten material is presented. The model used process parameters to obtain the temperature distribution, liquid and solid-state material transformation, and residual stress. The crack initiated by the thermal stress in the recast layer was investigated, and a mathematical model of crack tip stress was proposed. The results showed a wide range of residual stresses from 44 MPa to 404 MPa. The model is validated using experimental data with three points on the surface layer.

## 1. Introduction

SiCp/Al is a composite material made by adding silicon carbide particles with aluminum as a matrix and using specific technological means. This composite material has the characteristics of both aluminum and silicon carbide particles, such as electrical conductivity, low thermal expansion coefficient, low wear resistance, low corrosion resistance, etc., so it is widely used in precision fields such as aerospace, instrumentation, and optical components. However, due to the inhomogeneity of SiCp/Al composites, and containing a certain proportion of non-metallic particles with high strength, high hardness and high wear resistance, it is very difficult to process such materials by traditional machining methods. The main problems are as follows: The material processing is difficult and the processing quality is low. Especially during machining, the rake face is affected by the friction of the silicon carbide particles, resulting in severe tool wear, accumulation of debris, and low machining efficiency, making it challenging to meet machining requirements. Electrical discharge machining (EDM) relies on the spark discharge between two electrodes to remove the material. It is not limited by the strength and hardness of the material, and there is no physical contact between the tool and the workpiece during machining. Therefore, EDM is suitable for the machining of SiCp/Al composite materials.

Many scholars have conducted experiments on residual stress. Li et al. [1] experimented investigation on Cr12MoV Steel. The result showed that the depth of residual stress generated is much the same with different working liquids, but the working liquid with worse cooling capacity causes greater residual stress. Liu et al. [2] found the maximum value of average residual stress in the subsurface instead of the top surface owing to the high surface roughness. Pujari et al. [3] studied wire EDM parameters on residual stresses in the machining of aluminum alloy. The results obtained showed a wide range of residual stresses from 8.2 to 405.6 MPa. Mehmood et al. [4] found that the amount of residual stresses is proportional to the discharge current near the surface up to the depth of 75 µm. Ramulu et al. [5] studied the surface quality and subsequent performance of a 15% SiC particulate reinforced A356 Al. The results showed that the fatigue strength and residual stress of the material were significantly reduced using the tool with good conductivity. Batish et al. [6] analyzed the effect of parameters that induced residual stresses during electrical discharge machining of SiCp/Al. The pulse-off time was identified as the most significant factor in the formation of residual stresses. Additionally, better conductive electrode materials used during machining cause lower residual stress. Sidhu et al. [7] studied the influence of discharge parameters on the residual stress of the machined surface in EDM of SiCp/Al. The result showed that the workpiece, tool material properties, and pulse off time significantly contributed to the formation of residual stress. The concentration of reinforced particulates and matrix conductivity also play a vital role in the development of residual stress.

Many scholars have studied the changes of residual stress on the surface of SiCp/Al after EDM machining by experimental means. Still, few scholars have explored the change process of residual stress in the solidification process of molten material by both means of simulation and experiment. This paper linked the flow field with the structural field to simulate the change of residual stress during heating and solidification of SiCp/Al composites. The residual stress and the crack distribution were obtained. The discharge experiment was carried out on a self-built EDM machine to validate the simulation results, and the residual stress was measured by PROTO LXRD 3000.

## 2. Analysis of Residual Stress and Crack Formation

The solidification process of molten metal and molten particles mainly occurs in three phases. The first is that the molten material thrown into the working liquid solidifies when the working liquid cools and forms the debris. The second is that part of the molten material is thrown onto the corresponding electrode surface and cooled. Third, the residual materials that have not been thrown out of the pit cooled on the bottom of the pit. When new working liquid is introduced, a recast layer is formed on the surface, as shown in Figure 1.

The residual stresses are generated near the surface of the pit on both the tool and the workpiece, which mainly come from two aspects. Firstly, the heat source is the most critical factor affecting residual stress of discharge. The heat sources in EDM come from the discharge channels generated between the electrodes. When the heat source acts on the surface of the two electrodes, the heat on the surface of the tool and the workpiece promotes the melting and vaporization of the material by consuming its energy. In this process, a small amount of heat is still transferred to the tool and the workpiece. At this point, the physical properties of the material, such as the coefficient of expansion, will cause the material to expand and generate thermal stress [8]. For particle reinforced metal matrix composites, the coefficient of thermal expansion of matrix is different from that of particle. As the temperature rises, the internal stresses are released. Due to the difference in thermal expansion coefficient between the matrix material and the particles, the expansion of the matrix is much larger than that of the particles. In the expansion process, the matrix and the particles will inhibit each other and generate compressive stress, respectively. To relieve the stress, local microplastic deformation occurs in the material, and the stress below the yield strength is retained inside the workpiece as residual thermal stress [9,10].

The second is the phase transition stress caused by the material phase transition during rapid heating and cooling. As the temperature of the material decreases during cooling, the distance between the molecules will shorten, making the volume shrink. When the material is lower than its melting point, it will cause a phase transition from liquid to solid. The distance between molecules will be shortened again, generating phase transition stress [11].

Due to impurities, pores, and other defects as well as heating and cooling reasons, the workpiece will have an uneven distribution of stress during and after EDM machining. When the stress value exceeds the yield strength limit of the material, the microcrack forms. At this point, stress concentration occurs in the crack tip area. Although crack propagation can alleviate local stress, the stress concentration at the crack tip will lead to further crack propagation in both length and width, resulting in macroscopic crack formation.

## 3. Mathematical Model of Crack Tip Stress

When the residual stress is greater than the yield strength limit of the material, to alleviate the stress concentration, the local microplastic deformation of the material occurs. In this process, it is easy to promote the formation and expansion of the internal microcracks of the material. Generally, surface cracks are in opening mode after EDM machining [12]. Therefore, a crack model was established to analyze the residual stress at the crack tip, as shown in Figure 2.

Select the minimum area of the pit surface. Firstly, a YOZ coordinate system was established in the plane to analyze the crack propagation, and the crack was placed in the negative direction of the Z-axis. Therefore, local boundary conditions on the pit surface were determined as follows:(1)$$\begin{array}{l} y=0,-a<z<0,\sigma_{y}=0\\ y=0,z\to -\infty ,\sigma_{y}\to \sigma \\ y=0,z=-a,\sigma_{y}\to \infty \end{array}$$
where $\sigma_{y}$ is the stress in the y-direction. Functions of a complex variable are usually used to solve the plane crack stress problem, and Westergaard is a common solution. The stress function is selected as:(2)Ψ=Reϕ=I(ξ)+y Im ϕ¯I(ξ)
(3)σx=ReϕI(ξ)−y Im ϕI′(ξ)
(4)σy=ReϕI(ξ)+y Im ϕI′(ξ)
(5)$$\phi_{I}(\xi )=\frac{\sigma }{\sqrt{1+\frac{a}{\xi }}}$$
where Re represents the real part of functions of a complex; Im represents the imaginary part of functions of a complex; $\phi_{I}\left(\xi \right)$ is a widely used analytical function, ${\bar{\phi }}_{I}\left(\xi \right)$ and ${\overset{=}{\phi }}_{I}\left(\xi \right)$ are the primary and secondary integrals, respectively; ξ = *z* + i*y*.

The origin is moved to the crack tip, and the complex number is represented by trigonometric functions. Then, a point *ξ* (*z*, *y*) on the plane is transformed into *η* (*r*, *ω*), so *y* = *r*∙sin*ω*, *z* + *a* = *r*∙cos*ω*, then use the trigonometric function to express the complex number on the plane as follows:(6)η=ξ+a

The radius of stress field near the tip *r* → 0, therefore $\eta =re^{i\omega }\to 0$, *ξ* = η−a→−a. $\sigma_{y}$ and $\sigma_{x}$ are expressed as follows:(7)$$\sigma_{x}=\frac{\sigma \sqrt{a\pi }}{\sqrt{r\pi }}\cos\frac{\omega }{2}(1-\sin\frac{\omega }{2}\sin\frac{3}{2}\omega )$$
(8)$$\sigma_{y}=\frac{\sigma \sqrt{a\pi }}{\sqrt{\pi r}}\cos\frac{\omega }{2}(1+\sin\frac{\omega }{2}\sin\frac{3}{2}\omega )$$

Let $K_{I}=\sigma \sqrt{\pi a}$, where $K_{I}$ is called the fracture intensity factor, and the expression between stress and fracture intensity factor can be obtained as follows:(9)$$\sigma_{y}=\frac{K_{I}}{\sqrt{\pi r}}\cos\frac{\omega }{2}(1+\sin\frac{\omega }{2}\sin\frac{3}{2}\omega )$$

According to the stress intensity criterion of fracture mechanics theory, when the stress intensity factor $K_{I}$ reaches the material fracture toughness $K_{\mathrm{IC}}$, the material fractures. Suppose the fracture intensity factor of Al alloy is $K_{\mathrm{IC}}$, the critical stress $\sigma_{\mathrm{ycr}}$ at fracture can be obtained as follows:(10)$$\sigma_{\mathrm{ycr}}=\frac{K_{\mathrm{IC}}}{\sqrt{\pi r}}\cos\frac{\omega }{2}(1+\sin\frac{\omega }{2}\sin\frac{3}{2}\omega )$$

## 4. Modeling of Melting, Throwing, and Solidification of Materials

### 4.1. Simulation Model of Melting and Throwing Processes

Figure 3 is the boundary conditions in the melting simulation model. The flow field geometric model was a rectangle with the length and height of 300 μm and 330 μm, respectively. The geometry models of tool, workpiece, and discharge channel were established in the flow field. The tool and workpiece were rectangular with a height of 100 μm, and the discharge channel height was 30 μm. To improve the calculation efficiency, a two-dimensional axisymmetric model was adopted in the simulation, and the left side of the flow field was set as the axisymmetric. Interface thermal resistance was selected for the contact surface between matrix and particles. The upper and lower sides and the right side of the flow field were set as walls. The materials and machining parameters are listed in Table 1. The tool was red copper, the workpiece was SiCp/Al, and the deionized water was used in the working liquid. DEFINE_PROFILE module was used to load the Gaussian heat flux function onto the tool and workpiece surfaces. The melting and solidification model was selected to simulate the melting and phase transition process of the material surface. The pulse-on was 20 μs in the simulation for single discharge in EDM. Because the simulation research focuses on the material surface temperature and the formation of the molten pool, the discharge area was finely meshed [13,14] in the mesh section, and the other area was coarsely meshed. The meshed model is shown in Figure 4. The generated mesh file was imported into Fluent software, boundary conditions and material properties were set, and after the simulation, the changes of the surface temperature and molten pool with time were observed.

The material throwing simulation was obtained by modifying boundary conditions based on melting results. The boundary between the lower surface of the tool and the upper surface of the workpiece was modified to interface, and the upper and lower surfaces of the discharge channel were set as velocity outlets. The boundary conditions in the throwing simulation model are shown in Figure 5.

### 4.2. Modeling of Solidification Processes

The material solidification model was established by extracting the geometric model and modifying the boundary conditions based on the simulation results of the material throwing. The approach is as follows: Extract the pit surface contour curve from the throw simulation result file, output all points of the curve to the text file, and then import the file into ICEM to re-fit and draw a new two-dimensional pit model, as shown in Figure 6. The pit surface and other surfaces of the workpiece were set as the wall surface, the discharge channel was set as the velocity outlet, and the top of the flow field was set as the pressure outlet. Because the simulation time of the solidification process was long, a large time step was adopted in the simulation to shorten the calculation time.

### 4.3. Modeling of Residual Stress on Pit Surface

As a fluid simulation software, the latest version of Fluent is able to calculate the stress–strain field. Still, due to its weak analysis of structural mechanics, it can only calculate the linear stress–strain, which cannot meet the requirements of calculating the residual stress during the solidification process. In the simulation, the method of one-way fluid-solid coupling [15] was used to calculate the residual stress in the recast layer and the stress distribution at the crack tip after solidification. The flow field calculation results were taken as the initial conditions and introduced into the structural field for stress solution analysis. First, the flow field and structure field modules in Workbench were established. The flow field module calculated the melting, throwing, and solidification process of the material, and the structure field module calculated the residual stress and strain in the recast layer.

To study the influence of residual stress on crack formation, it is necessary to modify the workpiece geometry and add preset cracks on its surface, as shown in Figure 7a. The preset crack geometry model was re-meshed to study the stress at different crack tips. In the simulation, the main research object was the residual stress at the crack tip. To obtain more accurate calculation results and higher calculation efficiency, the mesh at the crack tip is locally fined in this paper. In other non-important calculation areas, the mesh was coarse, as shown in Figure 7b. After the calculation in the flow field was completed, the results were loaded into the structure field. The pressure field and temperature field calculation results were read, and then they were attached to the pit surface as the initial conditions for solving the residual stress. Transient analysis was adopted, and finally, the simulation calculation was carried out.

## 5. Simulation Results and Analysis

### 5.1. Analysis of Material Melting Process

Figure 8 shows the temperature field distribution on the tool and workpiece. The calculation error is 0.1%, so the values in the figures are all rounded. In Figure 8a, the maximum surface temperature of the workpiece at 5 μs is close to 14,100 K. This is mainly because the high-energy plasma in the discharge channel moves towards the electrodes at a very high speed. When it is bombarded on the surface of the electrodes, the kinetic energy of the plasma is converted into heat energy, which instantly forms a heat source with a small diameter and high temperature. As the heat source has a short action time, the heat does not transfer to other areas in time, so the heat mainly concentrates near the discharge point. However, at this time, the maximum temperature of the tool surface is about 8900 K, which is much lower than the surface temperature of the workpiece. This is because the presence of interface thermal resistance reduces the thermal conductivity of SiCp/Al composite material. The thermal conductivity of copper is much higher than that of copper, and the heat can be rapidly transferred to other locations, with less heat accumulation and less temperature rise. Figure 8b is the temperature field at 20 μs. As the heat transfers, the central temperature drops to 11,100 K, and the highest temperature on the tool surface drops to 7600 K. It can be seen from the figure that as the discharge time increases, the maximum temperature gradually decreases.

Figure 9 shows the curve of the maximum surface temperature of the tool and workpiece with time. In general, the maximum surface temperature of SiCp/Al is higher than that of copper electrode. From the curve, the slope of the curve becomes smaller, indicating that with the increase of discharge time, the decrease of the temperature reduction rate becomes slow. It is shown that the temperature changes violently at the beginning in EDM, and then the rate of temperature change decreases gradually.

Figure 10a shows the molten pool at 5 μs. As can be seen from the figure, the molten pool is formed quickly at this time, and the material in the molten pool changes from the solid phase to liquid phase, and the radius is greater than the depth. This is because the heat generated by the discharge channel applied to the tool surface travels faster in the direction of a radius than in the direction of depth, so the radius of the temperature field is greater than the depth. When the material reaches the melting point, a phase transition occurs. The material changes from the solid phase to the liquid phase, and the metal matrix exists in the molten state locally, forming a molten pool. The melting point of SiC is 2300 K, which is much higher than that of Al, so when the Al matrix reaches the melting point, part of the SiC particles are still in the solid state. It can be seen from the figure that part of the SiC particles appears red, representing the liquid phase, while some SiC particles are blue, representing the solid phase. The SiC particles near the center of the workpiece surface have been completely melted, and those far away from the discharge center are still solid. In addition, there is also a melting pool on the surface of the tool. Compared with the melting pool of the workpiece, the volume of the melting pool on the tool is smaller than that of the workpiece, mainly because in the heating process, the thermal conductivity of the tool is greater than that of aluminum carbide, and the heat can be quickly transferred out. Hence, the local temperature is relatively low. In addition, the melting point of copper is 1350 K, which is much higher than that of Al, so the Al matrix melts first, and the molten pool volume is smaller than the workpiece. Figure 10b shows the molten pool at 20 μs. The pool increases significantly in depth and radius, mainly because the heat has transferred to the surrounding area, causing more material to melt.

### 5.2. Analysis of Material Solidification and Throwing Process

Figure 11 and Figure 12 show the temperature field and the phase transition in the solidification process in the discharge area of the molten material from 500 μs to 700 μs. At 500 μs, as can be seen from Figure 11a, the surface temperature of the pit is higher than the melting point of the matrix, and the metal mainly exists on the surface in liquid form. Most of the molten metal with SiC particles was thrown out of the matrix, and only a small part of the metal materials and SiC particles remain in the discharge area. This part of the material is subjected to continuous shear stress and remains on the pit surface during the throwing process [16]. At 600 μs, the maximum temperature of molten Al decreases, and the solidified layer appears gradually during the phase transition from liquid to solid state. As can be seen from Figure 12b, the solidification rate at the bottom center of the pit was significantly faster than that of other positions. With the increase in time, the movement speed of the SiC particles in the molten Al slows down. At 700 μs, the residual molten Al has completely solidified, and the SiC particles are also solidified on the surface of the workpiece. At this time, the recast layer is formed, and the material enters the cooling stage after solidification.

### 5.3. Analysis of the Residual Stress

The thermal expansion coefficient of particle and matrix is different. When the thermal stress exceeds the yield strength of the material, microplastic deformation occurs in local areas. Such microplastic deformation develops rapidly at the tip of the microcrack. The residual thermal stress below the yield strength of the matrix is preserved as the residual thermal stress. Figure 13 shows the equivalent stress field distribution after machining. It can be seen from the figure that the stress at the crack tip is relatively concentrated. As the temperature of the molten metal drops, the metal continues to solidify in the working fluid, and the state changes from liquid to solid. At this time, the distance between atoms is further shortened, and the volume of the metal is reduced. The material generates stress due to the shrinkage. The maximum stress in the figure is 404.49 MPa. The stress is mainly due to the phase transition stress during the transformation of the molten metal into solid metal. The result is close to the literature [17,18].

## 6. Surface Crack Observation Experiment

### 6.1. Experiment Setup

The experiment was carried out on a self-built EDM machine, as shown in Figure 14. 65 vol% SiCp/Al was selected as the workpiece, red copper with a diameter of 1 mm was selected as the tool, deionized water was used as the working liquid, RC power supply was used to provide discharge energy for EDM machining, and positive polarity machining was adopted. In the experiment, the discharge voltage of the power supply was 45 V, the peak current was 25 A, the discharge pulse width was 20 μs, and the discharge machining parameters were shown in Table 2. The surface of the tool and the workpiece were polished before machining to remove the oxide layer and other impurities. In the experiment, the single-pulse discharge was adopted. When a spark was generated between the tool and the workpiece, the control circuit immediately disconnected the power supply to realize a single effective discharge.

JSM6360-LV high pressure vacuum scanning electron microscope (HVSEM) was used to observe the surface crack distribution of the pits after machining. The specific parameters of SEM are shown in Table 3. To study the influence of residual stress on surface cracks, three positions on the pit surface were selected for observation after the experiment. They are the center position of the pit, the middle position of the pit along the radius, and the position of the edge of the pit. The three positions A, B, C are shown in Figure 15. FV1000 electron microscope was used to take the overall photo of the pit, as shown in Figure 16.

### 6.2. Experimental Result and Analysis

Figure 17 shows the distribution of cracks on the surface of the pit at different positions observed by SEM. Figure 17a is the surface morphology at position A. As seen from the figure, there is no visible crack in this area. This is mainly because the stress in the central area of the pit is very small, the stress does not exceed the yield limit of the material, and no plastic deformation occurs in this area. It is difficult for the microcracks to form visible macroscopic cracks. Therefore, no cracks are observed here. Figure 17b is the surface morphology at position B. Cracks can be observed in this region, and the crack inside the square has a slender shape. This is mainly because during the heating process, the materials constrain each other due to thermal expansion, causing local stress between the two materials. In general, the thermal expansion coefficient of the metal matrix is large, but particles are much smaller. The difference of thermal expansion coefficient results in thermal mismatch stress, which leads to microplastic deformation of the material. This plastic deformation develops rapidly at the tip of the microcrack, and then promotes the initiation and propagation of the microcrack. Figure 17c is the surface morphology at position C. The crack in the block is obvious and has a slender shape. This is because the stress concentration here is larger than the yield strength limit of the material itself, and the plastic deformation is larger than that at position B, so the crack is further expanded in length and width [19].

The main reason for crack propagation in Figure 17b,c is that extremely high temperature is generated instantly on the tool and workpiece surface during EDM, and the temperature decreases rapidly after discharge, which causes the material to expand and contract in a short period and be subjected to the thermal shock of this cycle. Thermal stress and thermal fatigue occur inside the material. Due to defects such as pores, impurities and microcracks, the stress concentration at the microcrack tip easily results in the formation of macroscopic cracks when subjected to thermal impact. In the simulation, the tip stresses at position B and position C on the surface of the pit are larger. In the experiment, large cracks appear at position B and position C, indicating that a large residual stress is generated inside the pit and is greater than the yield strength of the material, resulting in microscopic plastic deformation in the material. The crack location and size in the experiment can indirectly verify the distribution of residual stress obtained in the simulation, which verifies the correctness of the simulation results.

In the experiment, the X-ray diffractometer can only measure the surface of the workpiece. Proto LXRD 3000 was used to measure the residual stress of the three points A, B, and C. The maximum stress exists on Position B, which is 164 MPa. The second largest stress is on position C, 131 MPa. The minimum stress is 86 MPa, which is on position A. The largest stress normally appears at the crack tip. In the simulation, a small area for each position is selected, and the average value is calculated. The simulation result at position A is 90.25 MPa, the result at position B is 168.06 MPa, and the result at position C is 110.63 MPa. The results are close to those results in literature [6,7]. Compared with the simulation results, the experimental results are similar to the simulation results and verify the distribution of the cracks on the surface of the pit. The simulation and experimental results are shown in Figure 18.

After machining, to explore the elements on the surface, the spectrum analysis experiment was conducted. The impurities of the workpiece were removed by an ultrasonic cleaning machine. The types and contents of elements are shown in Table 4. The surface contains five elements, namely carbon, oxygen, aluminum, silicon, and copper. Oxygen comes from surface pollutants, while copper comes from the tool. When a discharge channel is established between the two electrodes, the electrode surface melts. Under the disturbance of working liquid, part of copper metal solidifies on the pit surface. There are complex chemical and physical changes in the discharge process, and new substances are produced. Some of these substances are free in the working fluid, and some are solidified on the surface of the pits to form a new recast layer.

## 7. Conclusions

In this paper, the one-way fluid-solid coupling method was adopted to establish the process of material melting, throwing, and solidification in the flow field. The residual material stress model in the recast layer was established in the structural field. The surface micromorphology and crack distribution state of the residual material after solidification were explored. The conclusions are as follows:(1)At the initial moment of discharge, the largest temperature of the tool is 13,532 K and the largest temperature of the workpiece is 15,896 K. With the increase of discharge time, the maximum surface temperatures of the two electrodes present a downward trend, and the molten pool areas continue to expand. As the solidification process progresses, a new recast layer is formed on the surface of the pit.(2)With the increase of time, the temperature of molten liquid on the surface of the pit decreases slowly. At 700 μs, the residual molten Al on the surface of the pit completed solidified. Meanwhile, the stress at the crack tip also decreases slowly. Micro-plastic deformation occurs in the material, and the residual stress is relaxed.(3)In the surface crack observation experiment, the crack grows gradually from the center of the pit to the edge, indicating that the stress increases gradually along the radius direction, promoting the initiation and further expansion of microscopic cracks and the formation of macroscopic cracks.(4)The smallest residual stress is 86 MPa and lies on the center of the pit. The highest stress is 164 MPa and lies in the middle of the bottom center to the edge of the pit, which is close to the simulation results.

In the modeling, the study conducted a one-way coupling fluid structure interaction method. In fact, during solidification, residual stresses are generated synchronously. In the future, the two-way coupling method will be considered in the simulation to improve the accuracy of the results.

## Figures and Tables

**Figure 1 micromachines-13-00972-f001:**
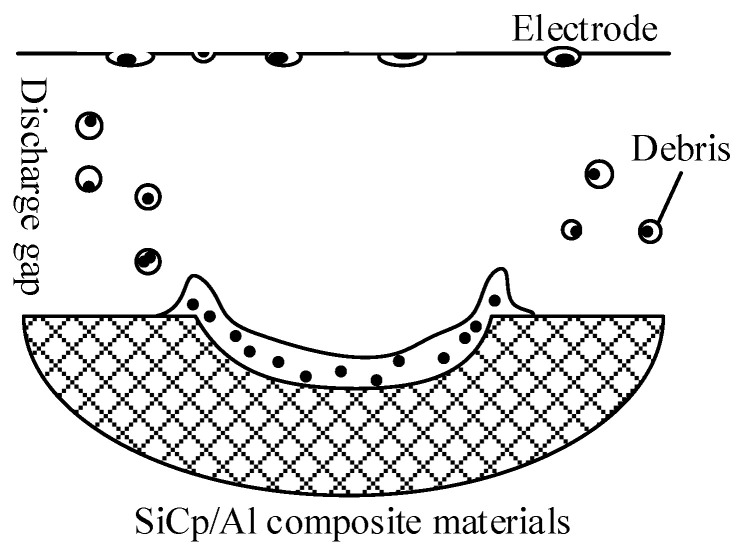
The process of solidification process for molten material.

**Figure 2 micromachines-13-00972-f002:**
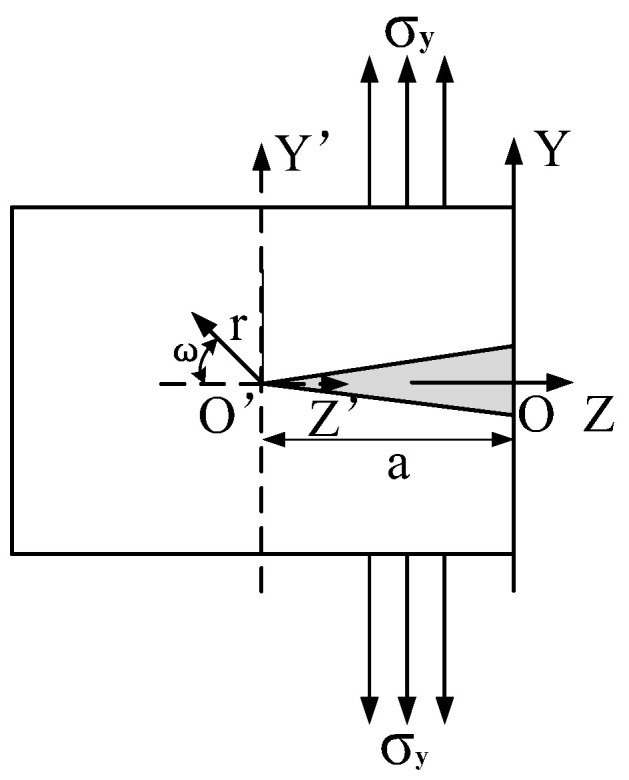
Schematic diagram of the crack model.

**Figure 3 micromachines-13-00972-f003:**
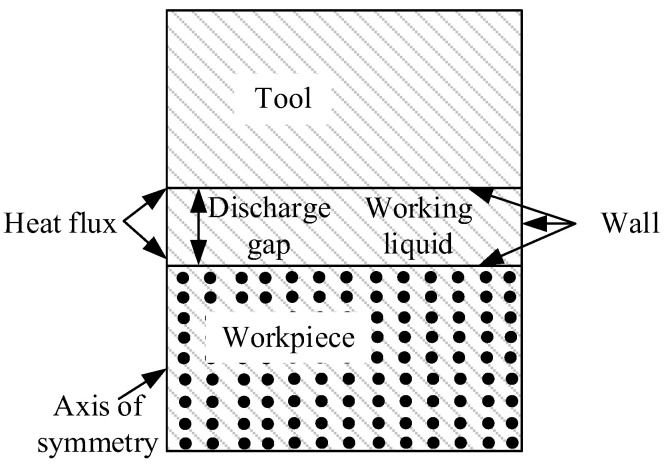
The boundary conditions in the melting simulation model.

**Figure 4 micromachines-13-00972-f004:**
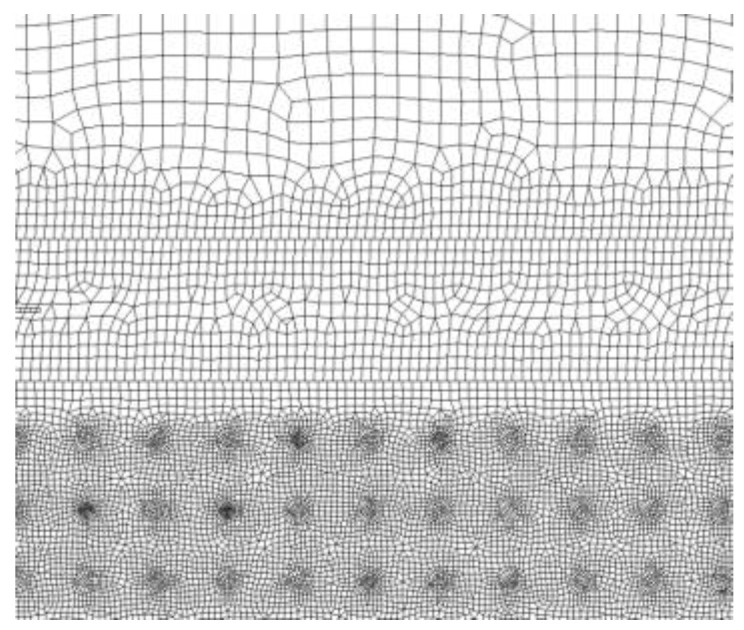
The meshed model.

**Figure 5 micromachines-13-00972-f005:**
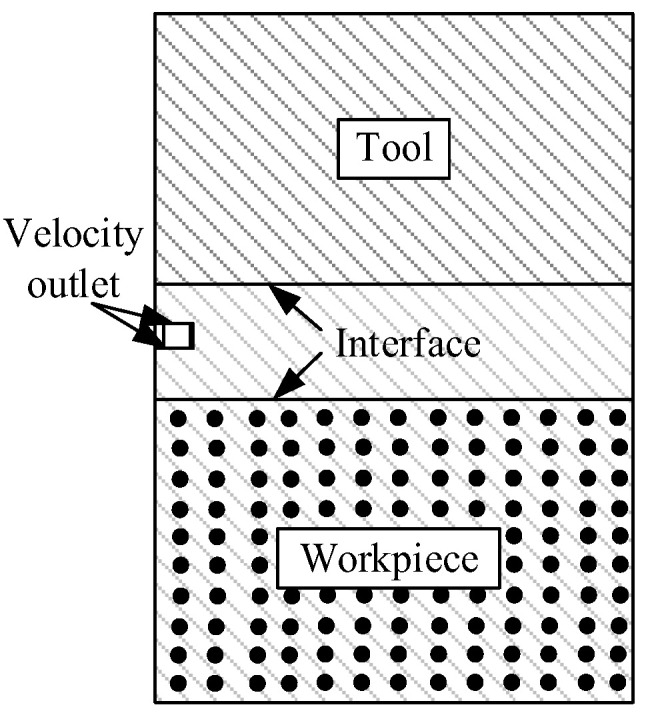
The boundary conditions in the throwing simulation model.

**Figure 6 micromachines-13-00972-f006:**
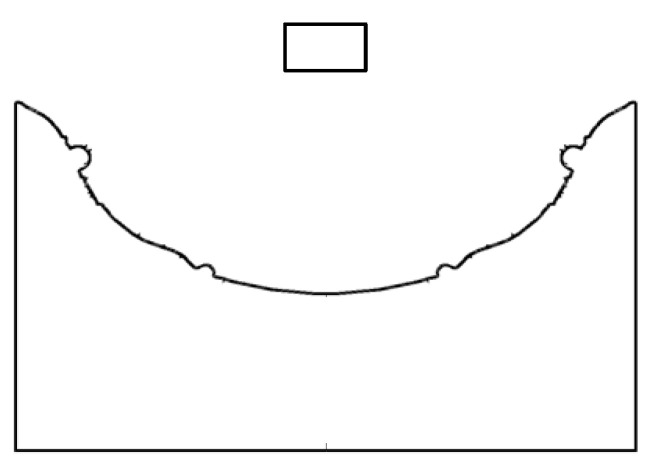
The 2D pit model after fitting.

**Figure 7 micromachines-13-00972-f007:**
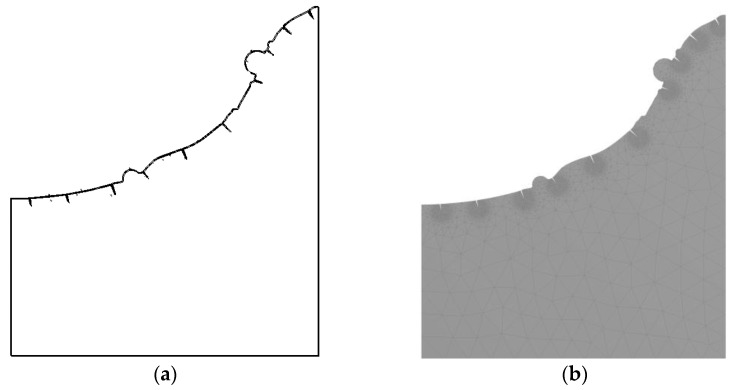
Geometric and meshed model of preset crack. (**a**) A geometric model of a preset crack; (**b**) meshed model.

**Figure 8 micromachines-13-00972-f008:**
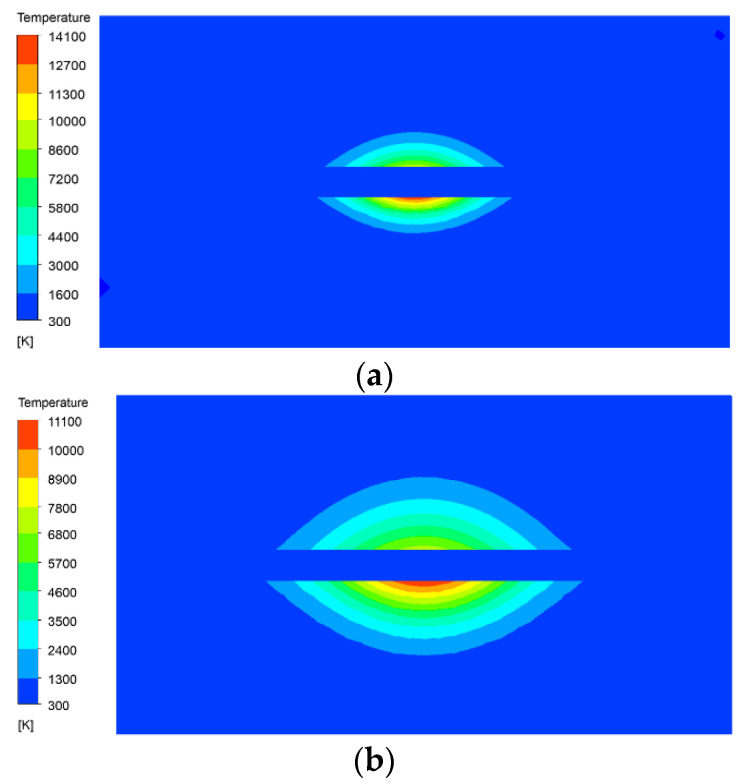
Temperature field distribution on the tool and workpiece at different discharge moments. (**a**) Discharge time at 5 μs; (**b**) Discharge time at 20 μs.

**Figure 9 micromachines-13-00972-f009:**
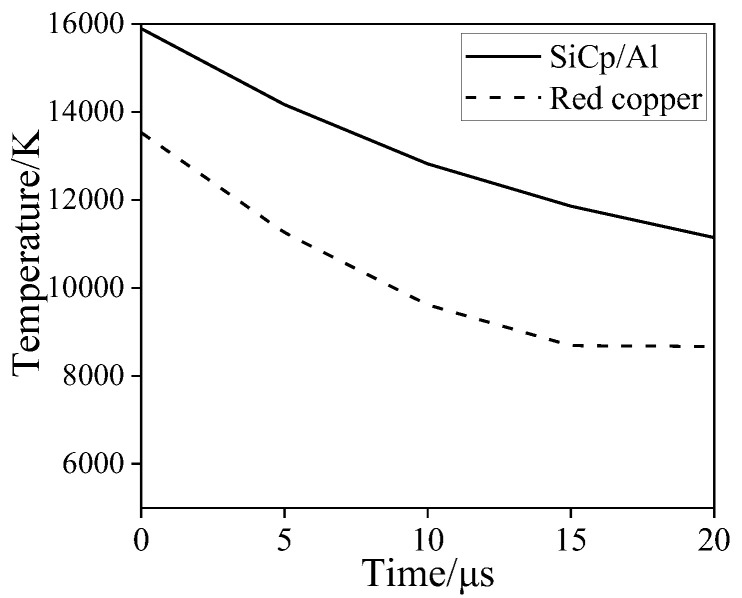
Curve of the maximum temperature of red copper and SiC_p_/Al with time.

**Figure 10 micromachines-13-00972-f010:**
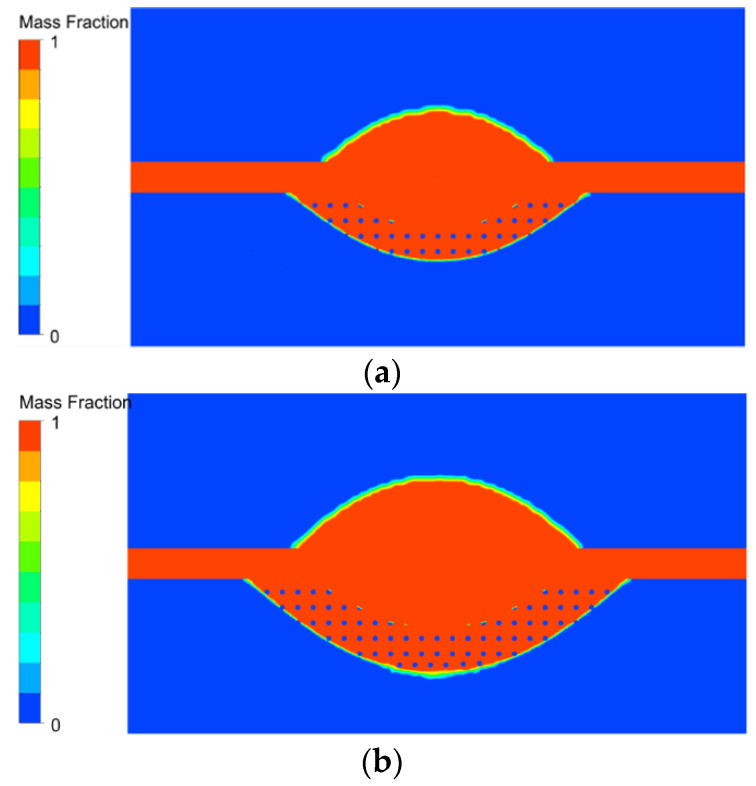
Molten pool on the tool and workpiece at different discharge moments. (**a**) Discharge time at 5 μs; (**b**) Discharge time at 20 μs.

**Figure 11 micromachines-13-00972-f011:**
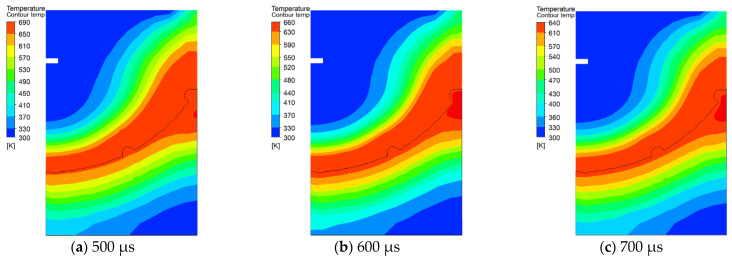
The temperature field changes at different moments.

**Figure 12 micromachines-13-00972-f012:**
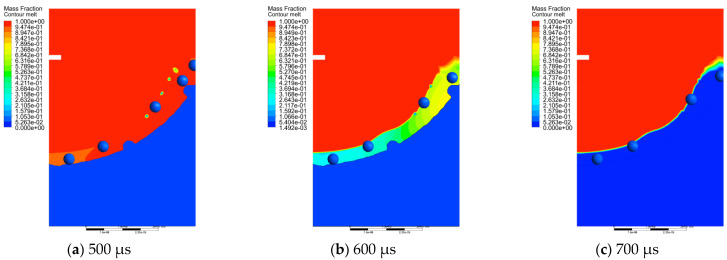
Phase of solidification process of molten metal at different moments.

**Figure 13 micromachines-13-00972-f013:**
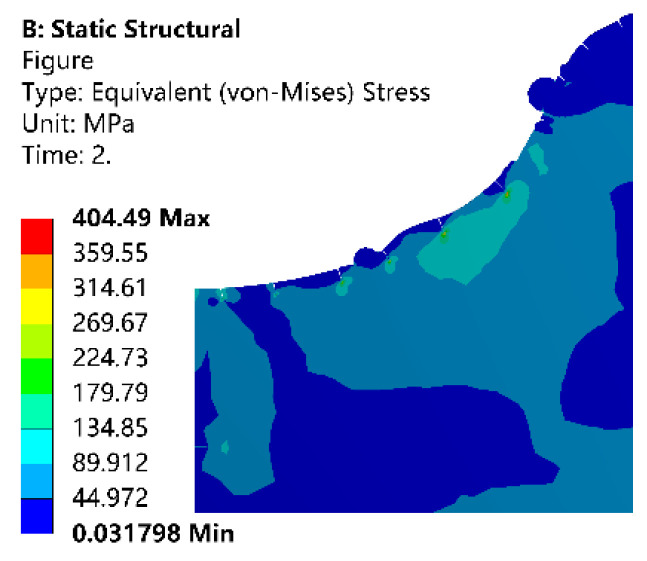
The residual stress after machining.

**Figure 14 micromachines-13-00972-f014:**
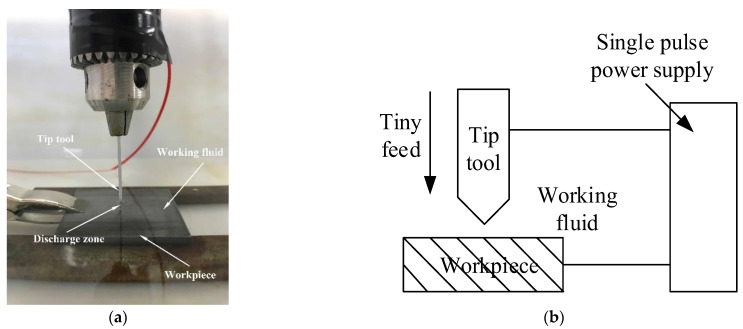
Experimental process of single pulse tip discharge and its schematic diagram. (**a**) Tip discharge with single pulse; (**b**) Tip discharge process.

**Figure 15 micromachines-13-00972-f015:**
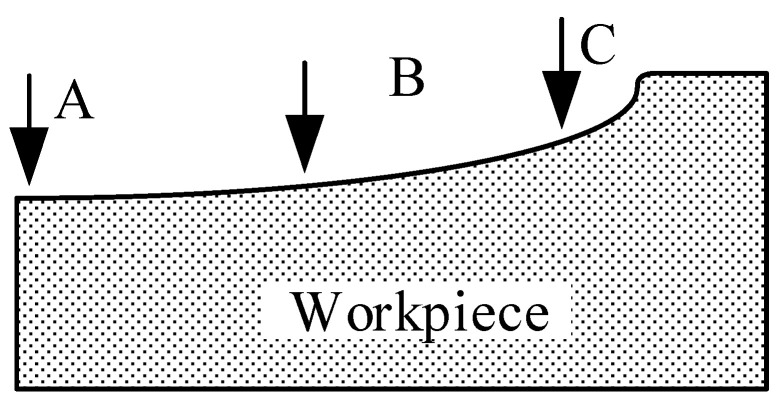
Positions on the surface of the pit.

**Figure 16 micromachines-13-00972-f016:**
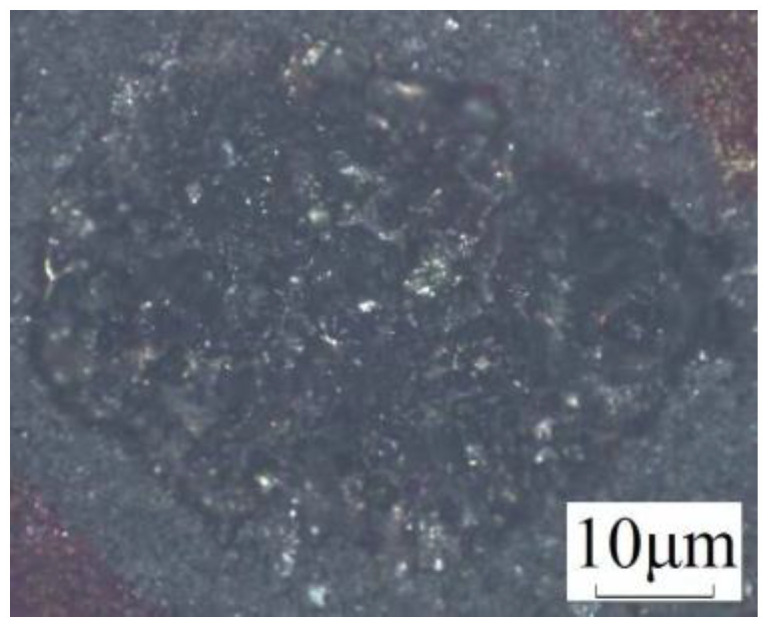
The overall photo of the pit.

**Figure 17 micromachines-13-00972-f017:**
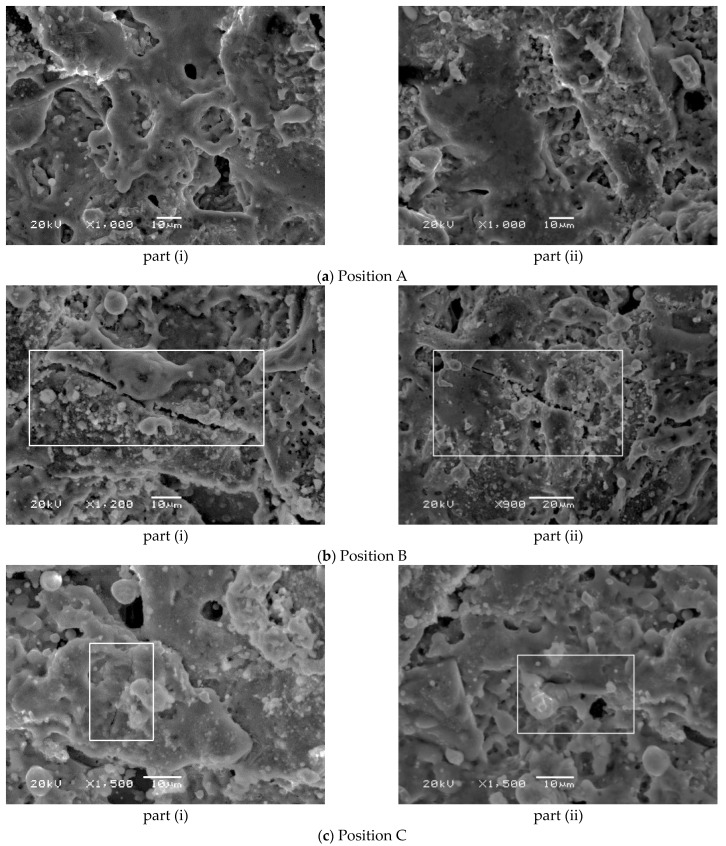
The cracks on the pit surface.

**Figure 18 micromachines-13-00972-f018:**
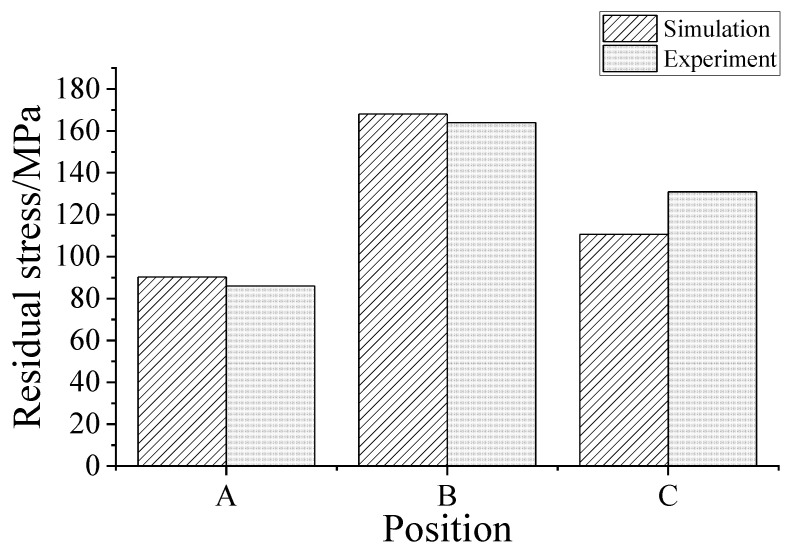
Residual stress at different positions.

**Table 1 micromachines-13-00972-t001:** Simulation parameters in the melting simulation.

Parameter	Description
Tool material	Red copper
Workpiece material	SiCp/Al
Working liquid	Deionized water
Voltage *U* (V)	45
Current *I* (A)	20
Pulse-on *T*_on_ (μs)	20
Particle diameter	5 μm
Energy distribution (tool)	0.23
Energy distribution (workpiece)	0.30

**Table 2 micromachines-13-00972-t002:** Discharge parameters in EDM machining.

Parameter	Description
Tool material	Red copper
Workpiece material	SiCp/Al
Working liquid	Deionized water
Tool diameter (mm)	1
Voltage *U* (V)	45
Current *I* (A)	25
Pulse-on *T*_on_ (μs)	20

**Table 3 micromachines-13-00972-t003:** The parameters of SEM.

Parameter	Description
LV pressure *P_L_* (Pa)	1~270
Accelerating voltage *V_a_* (kV)	0.3~30
Specimen stage *s* (mm)	X: 125, Y: 100, Z: 5–80
Tilt *R_l_* (°)	−10–+90
Rotation *R_t_* (°)	360
Magnification *n*	×5–×100,000

**Table 4 micromachines-13-00972-t004:** Elements and weights.

Elements	C	O	Al	Si	Cu
Contents (%)	6.25	18.08	14.13	16.58	44.96

## Data Availability

The data presented in this study are available on request from the corresponding author. The data are not publicly available due to further study.

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
