# Peer review of "Study on Solidification Process and Residual Stress of SiCp/Al Composites in EDM"

_micromachines, 2022, doi:10.3390/mi13060972_

Round 1

Reviewer 1 Report

The work is devoted to an actual problem. Metal matrix composites based on aluminum and silicon carbide are important materials. The study of their properties, synthesis processes, modeling of the processes associated with them are important scientific tasks.

The work needs to be improved; may be published after corrections.

There are some minor typos, but the text should be double-checked.

In the second paragraph of the introduction section, the names of the authors are listed and a description of their results is given, it seems to me that it would be more correct to indicate a bibliographic reference if this does not contradict the requirements of the journal.

In the first paragraph of the introduction, the problem is indicated by the word "However", while the explanation follows. The problem in its current form is not obvious, I recommend describing the problem a little differently and more clearly.

The simulation section describes the results, where it is said that the temperature is reached 14169 K; probably, there is a certain zone of the confidence interval of the calculation (inaccuracy), which should be indicated; in addition, measurements in thousands of degrees cannot be calculated with an accuracy of 1 degree (14169 K), values ​​in the text and figures should be rounded, and confidence intervals should be indicated.

Some graphs can be improved, for example, fig. 19 is shown practically without processing from the energy dispersive analysis data, it can be taken in the form of a data array, built, and there will be a more beautiful picture. And in general, 19 figures for the article, it seems to me that this is a lot, you can take the auxiliary data to the section of additional data.

In the list of references, a number of references are designed as a DOI index, not as a bibliographic reference; there must be uniformity and design with the requirements of the publishing house.

Reviewer 2 Report

In the manuscript, “Study on solidification process and residual stress of SiCp/Al composites in EDM”, Zhang and co-workers have reported the stress simulation of crack appearance during solidification process of SiCp/Al composites. Furthermore, authors compared their simulation results with experimental values. It’s a well written paper. I have minor criticisms and suggestion to improve the manuscript.

1.      Are there any other simulation studies of this kind? If yes, authors should compare results with the literature as well.

2.      Some figures can be improved; e.g., Fig. 10, 11, 12, 13 – side labels are not visible. Use better quality figures.  

3.      Figure 17, Caption should be written in detailed; labelling parts (i) and (ii) for each position. Explaining each part in caption itself.

4.      Again, Figure 19 labels are not clear. Part b, peaks are not readable.

5.      Line 209 “…tudy” I think, it’s a typo. It should be “To Study”. Please check the whole manuscript for typos and errors.

Round 2

Reviewer 1 Report

Notes have been corrected. The manuscript can be published.